# SAT3D: Image-driven Semantic Attribute Transfer in 3D

Submission Id: 2290*

## ABSTRACT

GAN-based image editing task aims at manipulating image attributes in the latent space of generative models. Most of the previous 2D and 3D-aware approaches mainly focus on editing attributes in images with ambiguous semantics or regions from a reference image, which fail to achieve photographic semantic attribute transfer, such as the beard from a photo of a man. In this paper, we propose an image-driven Semantic Attribute Transfer method in 3D (SAT3D) by editing semantic attributes from a reference image. For the proposed method, the exploration is conducted in the style space of a pre-trained 3D-aware StyleGAN-based generator by learning the correlations between semantic attributes and style code channels. For guidance, we associate each attribute with a set of phrase-based descriptor groups, and develop a Quantitative Measurement Module (QMM) to quantitatively describe the attribute characteristics in images based on descriptor groups, which leverages the image-text comprehension capability of CLIP. During the training process, the QMM is incorporated into attribute losses to calculate attribute similarity between images, guiding target semantic transferring and irrelevant semantics preserving. We present our 3D-aware attribute transfer results across multiple domains and also conduct comparisons with classical 2D image editing methods, demonstrating the effectiveness and customizability of our SAT3D.

## CCS CONCEPTS

• **Computing methodologies → Image manipulation**.

## KEYWORDS

Attribute Transfer, Image Editing, 3D

## 1 INTRODUCTION

The objective of GAN-based image editing task is to alter particular attributes of an image by manipulating the corresponding latent codes within the generative model's latent space [26]. With image editing technologies, people can modify their facial expressions in photos without reshooting, merchants can guide consumers through virtual hair customization and clothing try-on, and designers can edit their ideas more efficiently in a real-world scenario. Therefore, image editing has been a crucial task in the computer vision and multimedia community.

**Figure 1: Comparison of different image editing tasks. Existing methods focus on editing with ambiguous semantics or regions from images. Generally, the semantic attributes are described by texts or classifiers, suffering from ambiguity. Illustrating specific characteristics of attributes with reference images can clarify descriptions. The existing image-driven methods are based on region-wide replacement, which are unable to migrate semantic attributes, such as beards. Instead, our proposed SAT3D is an image-driven semantic-based method, enabling the editing of semantic attributes according to the details of reference images.**

The existing 2D and 3D-aware image editing methods, from classical StyleGAN-based networks to emerging diffusion-based networks, can be broadly categorized into semantic-driven and image-driven region-based approaches, as presented in Figure 1. Since texts and classifiers cannot accurately and comprehensively characterize an attribute, semantic-driven methods guided by texts [4, 28] or classifiers [32] usually suffer from vague descriptions and limited customizability. Conversely, the image-driven region-based methods [33, 39] can migrate specific characteristics of the target region from reference images onto the source image. However, such migrations are performed on the entire region and unable to distinguish between multiple semantic attributes within the same region, e.g., a face region includes semantic attributes such as skin color, beard, and so on. These drawbacks motivate us to design a method for semantic attribute transfer based on reference images with 3D-aware ability.

In this paper, we propose our image-driven attribute transfer method SAT3D by learning correlations between semantic attributes and style code channels of a pre-trained 3D-aware generative model. Since the semantic attributes are defined with phrase-based texts,

SAT3D is able to distinguish different attributes in the same region. The difference between the style code of source and reference images on relevant channels reveals the editing direction for the target attributes, enabling highly customizable attribute editing based on the detailed characteristics from reference image. For training, we define a set of phrase-based descriptor groups for each attribute, and develop a Quantitative Measurement Module (QMM) to quantitatively describe the attribute characteristics in images based on the descriptor groups, leveraging the image-text comprehension capability of Contrastive Language-Image Pre-training (CLIP) model [29]. Then we design a target attribute transfer loss and an irrelevant attribute preservation loss based on QMM for the edited images, guiding the migration of target attributes towards reference images while minimizing alterations to others. Notably, SAT3D also can deal with 2D image editing effectively by adopting a 2D StyleGAN-based generator. The main contributions can be summarized as follows:

- We propose an image-driven semantic attribute transfer method in 3D termed SAT3D based on the pre-trained 3D-aware generator. Notably, with the generalizability, SAT3D is also applicable with other 2D and 3D-aware StyleGAN-based generators.
- We develop a Quantitative Measurement Module (QMM) to quantitatively describe the attribute characteristics in images, and design specific training losses for target attribute transfer and irrelevant attributes preservation.
- We provide 3D-aware attribute transfer results across multiple domains and perform comparisons with classical 2D image editing methods, demonstrating the effectiveness and customizability of SAT3D. To the best of our knowledge, this is the first work for image-driven semantic attribute transfer task in 3D.

## 2 RELATED WORKS

**Generative Image Synthesis.** The field of 2D image generation has been extensively and intensively explored. Early variants of GAN networks [10, 12, 22, 30] take latent code directly as input, resulting in the coupling of high-dimensional and low-dimensional features. StyleGAN series models [17–20] use mapping networks to decouple the latent code and map it to style parameters through affine transformations, allowing for style migration and fusion by disentangling features at different dimensions.

Recently, generative 3D-aware image synthesis is also drawing more attention. Early voxel-based 3D scene generation methods are limited to low resolution due to high memory requirements. With the explosion of neural rendering methods, e.g., NeRF [27], fully implicit 3D networks [6, 31] are proposed, but still with a high query overhead, limiting training efficiency and rendering resolution. To achieve high-resolution 3D-aware image synthesis, StyleNeRF [11] and CIPS-3D [43] render features instead of RGB colors based on NeRF implicit representations. While EG3D [5] and StyleGAN3D [42] use hybrid tri-plane representation and MPI-like representation respectively based on StyleGANv2, achieving high quality rendering in a more computationally efficient way. We implement 3D-aware attribute transfer based on the pre-trained EG3D generator, which improves efficiency with hybrid tri-plane representation and provides disentangled latent space with StyleGANv2 architecture.

**Attribute Transfer.** There are a number of approaches for attribute editing and transferring, which generally fall into two categories: semantic-driven methods and image-driven region-based methods. The semantic-driven methods manipulate latent codes guided by text [1, 15, 24, 25, 28] and attribute classifiers [2, 13, 14, 32, 34]. Although effective, these descriptors are ambiguous in nature, leading to randomness in the generated results. In contrast, our approach uses image samples as reference to specify attribute features, which reduces ambiguity and the inconvenience to users.

Another group of attribute editing methods perform image-guided multi-attribute transfer based on regions. SEAN [44] distills per-region style codes from the reference image based on semantic mask, and controls target attributes with style codes and a mask jointly, which is relatively inconvenient for users. SemanticStyleGAN [33] and StyleFusion [16] enable latent codes to represent different regions in a disentangled way, and then achieves attribute transfer by directly replacing latent codes of the corresponding region. However, the region-based disentanglement in these methods cannot distinguish between different attributes with overlapping regions, such as beard on the face, whereas our proposed SAT3D selects channels based on textual semantics, providing greater flexibility. In terms of efficiency, our approach is based on pre-trained generators without requiring restructuring of latent spaces or fine-tuning of models, which greatly saves training time and resources.

With the recent explosion of diffusion models, there are also increasing works investigating diffusion-based image editing, but these works are similarly categorized into semantic-driven methods [3, 4, 21, 41] and image-driven region-based methods [39] as StyleGAN-based works, suffering from ambiguity and indivisibility. Moreover, as stated in [37], although the latent space of stable diffusion model shows the capability of disentanglement, it performs better for integral attribute editing but poor for local editing such as hair color, thus not suitable for fine-grained attribute transfer of facial images.

## 3 METHODS

Our SAT3D is proposed to address the image-driven semantic attribute transfer task, which aims at migrating specific attribute characteristics from a reference image onto the source image while suppressing irrelevant variations. The migration is not a regional replacement (i.e., face region), but based on descriptive semantic attributes (i.e., beard). The pipeline of SAT3D is presented in Figure 2. Leveraging the high disentanglement within the style space of pre-trained EG3D generator, we explore the relevant style code channels for each semantic attribute and identify the editing direction for migration accordingly (Section 3.1). The exploration is guided by our designed attribute loss function, which utilizes the zero-shot prediction ability of pre-trained CLIP models to quantitatively measure the attribute characteristics in images based on the descriptor groups (Section 3.2).

### 3.1 Semantic Attribute Channel Discovery

**Latent Space.** EG3D extends StyleGAN-based image generation to geometry-aware multi-view generation, comprising similar latent spaces $\mathcal{Z}$, $\mathcal{W}$, and $\mathcal{S}$. For EG3D, given a random latent code $z \in \mathcal{Z}$ and a camera pose matrix $\mathbf{P}$, the mapping network transforms

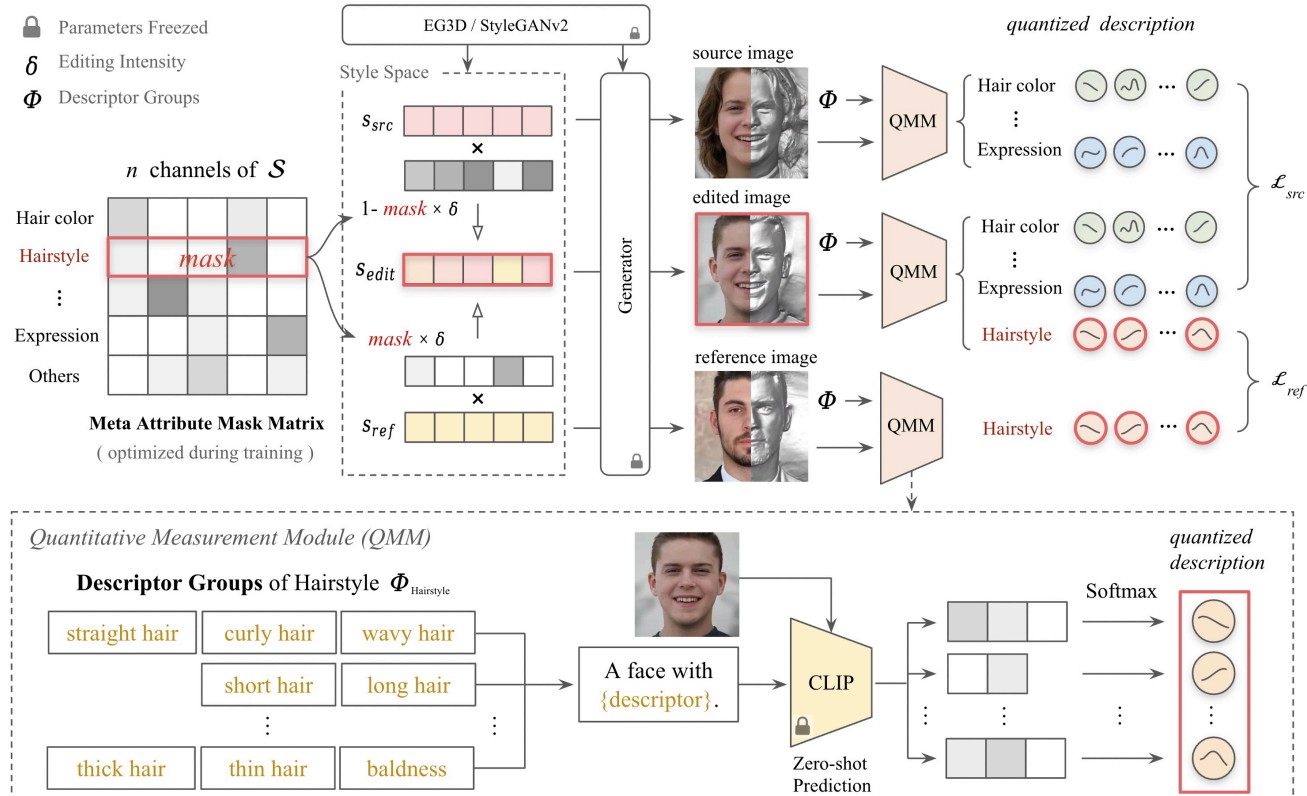

**Figure 2: The attribute transfer pipeline of SAT3D. Based on pre-trained 2D or 3D-aware generators, SAT3D learns a meta attribute mask matrix to explore correlations between semantic attributes and style code channels of style space $\mathcal{S}$. For training guidance, we define a set of descriptor groups $\Omega$ for each attribute and develop a Quantitative Measurement Module (QMM) to measure the attribute characteristics in images, utilizing the zero-shot prediction capability of CLIP. With QMM, the attribute losses are designed for target attribute transfer and irrelevant attribute preservation. For this example, the target attribute set $\Omega = \{Hairstyle\}$ and the editing intensity along editing direction $\delta = 1$. A lock indicates the module parameters being frozen.**

$[z, \mathbf{P}]$ to intermediate latent space $\mathcal{W}$. Then the different affine transforms in each layer of the generator further transform latent code $w \in \mathcal{W}$ to style code $s \in \mathbb{R}^n$ in $\mathcal{S}$ space, which performs better in disentanglement and completeness [38]. The space $\mathcal{S}$ is suitable for fine-grained attribute manipulation and transfer.

**Latent Manipulation.** Considering the disentanglement of style space in controlling different attributes, we can intuitively replace the attribute-related channels to transfer attributes between images, and then the challenge becomes how to explore the correlations between attributes and style code channels. Assuming there exist $m$ attributes, a Meta Attribute Mask Matrix $\mathcal{M} \in \mathbb{R}^{(m+1) \times n}$ which needs to be optimized is defined to represent correlations between $m$ attributes (plus an *others* item indicating exclusionary factors) and $n$ style code channels. For the task of transferring a target attribute set $\Omega$ of the reference image $I_{ref}$ to the source image $I_{src}$, $\mathcal{M}$ is first normalized with softmax along attribute dimension to produce a control probability distribution on attributes for each channel. Then the control probabilities of $\Omega$ for all channels are selected and summed along attribute dimension to produce a $mask \in \mathbb{R}^{1 \times n}$ for $s_{src}$ and $s_{ref}$. The edited style code can be merged by $s_{edit} = s_{src}(1 - mask) + s_{ref} mask$.

However, as found in practice, merely interpolating between the two style codes gives insufficient alteration strength to the source image in many cases. Thus we extract the editing direction from the source-reference image pair and allow users to adjust the editing intensity beyond [0,1], which provides a more comprehensive and customizable attribute transfer capability. Given that our method finds the attribute editing direction $(s_{ref} - s_{src}) \times mask$ over the source-reference image pair, and the editing intensity $\delta$ in the direction varies between different image pairs, the final edited style code is parameterized as $s_{edit} = (s_{ref} - s_{src}) \times mask \times \delta$. During the training process, $\delta$ is default set as 1, and it can be increased appropriately according to the image pair during inference.

**Channel Pre-selection for Complex Attributes.** While for complex attributes, e.g., hairstyle, there are hundreds of channels in $\mathcal{S}$ space taking control of different specific characteristics, which makes it inefficient to detect all related channels from scratch. Taking this into consideration, we pre-select the most activated channels by analyzing gradients from specified regions [36] when initializing $\mathcal{M}$. In each iteration during training, an image is generated by the pre-trained generative model from a randomly sampled pair of latent code $z$ and camera pose matrix $P$. Then for each semantic

region, the binary mask predicted by [40] is set as gradient maps of the generated image and the gradients in valid areas are back-propagated to corresponding style code channels. The absolute gradient values for each channel-region pair are recorded and normalized by region size. The averaged gradient values of the whole training procedure are calculated and normalized along semantic region dimension. For the attribute $t$ relating to region $r$ that needs pre-selecting, we sort the obtained values for $r$ in descending order and pick out $k_t$ channels with the largest value as the pre-selected style code channels. In this way, the network can learn to select relevant channels for easy attributes, and filter irrelevant ones from pre-selected channels for complex attributes. Both of the two approaches can be achieved efficiently.

During initialization, $\mathcal{M}$ is assigned all zeroes, and then for each channel $i$, $\mathcal{M}_t^i$ is set to $d_t$ if it is pre-selected by attribute $t$, while otherwise $\mathcal{M}_{others}^i$ is set to 1. Notably, the *toRGB* layers in StyleGANv2 architecture are used to convert feature maps to three-channel RGB images, and the *Super-Resolution* module in EG3D is designed to increase the image resolution. The style parameters in *toRGB* layers and the *Super-Resolution* module do not control the generation of image content, thus we exclude the corresponding style code channels in all procedures.

## 3.2 Learning Objective

To achieve optimal semantic attribute migration, we expect to migrate the target attributes while suppressing alteration of irrelevant attributes. To this end, a series of loss functions are proposed as guidance, i.e., attribute loss $\mathcal{L}_{attr}$ for attribute transfer and preservation, background loss $\mathcal{L}_{bg}$ for further variation suppression in uninterested regions, and probability loss $\mathcal{L}_{prob}$ for correlation focusing of each channel. The details are described as follows.

**Attribute Loss.** To describe the attributes conveniently and efficiently, we define a set of descriptor groups $\Phi_t$ for each attribute $t$ and develop a Quantitative Measurement Module (QMM), which takes advantage of the zero-shot prediction capability of CLIP [29]. Specifically, we utilize a toy example to introduce how does QMM quantitatively measure the attributes. As exemplified in Figure 2, each descriptor group consists of a set of phrases (with a text template 'a face with {}') describing a particular characteristic of the attribute, which can be considered as class labels of a classifier. Taking a phrase-image pair as input, CLIP yields a correlation score. Then for all the phrases in a descriptor group $\phi \in \Phi_t$, a correlation vector $CLIP(I, \phi)$ can be obtained. We normalize the correlation vector with softmax to produce classification probability $D(I, \phi) = softmax(CLIP(I, \phi))$ and regard $D$ as the metric to describe the characteristic of the attribute quantitatively.

Then to transfer a target attribute set $\Omega$, the attribute loss $\mathcal{L}_{attr}$ comprising target attribute transfer loss $\mathcal{L}_{ref}$ and irrelevant attribute preservation loss $\mathcal{L}_{src}$ is calculated as follows:

$$\mathcal{L}_{ref} = \sum_{t \in \Omega} \sum_{\phi \in \Phi_t} |D(I_{edit}, \phi) - D(I_{ref}, \phi)|, \qquad (1)$$

$$\mathcal{L}_{src} = \sum_{t \notin \Omega} \sum_{\phi \in \Phi_t} |D(I_{edit}, \phi) - D(I_{src}, \phi)|, \qquad (2)$$

$$\mathcal{L}_{attr} = \mathcal{L}_{ref} + \mathcal{L}_{src}. \qquad (3)$$

That is, for the edited image, the characteristics of target attributes are expected to be similar to the reference image guided by $\mathcal{L}_{ref}$, while other attributes should remain the same as the source image guided by $\mathcal{L}_{src}$.

**Background Loss.** In addition to the attribute preservation loss, we also impose a penalty on background change to prevent channels that control irrelevant factors from being selected. Specifically, we define an alterable semantic region for each attribute, e.g., the beard attribute corresponds to the bottom half of the face region. Denoting the predicted binary attribute mask as $B_{src}$ and $B_{edit}$ for the source and edited images respectively (1 for the alterable semantic region, 0 for others), the background mask $B$ and loss function is calculated by:

$$B = \overline{B}_{src} \& \overline{B}_{edit}, \qquad (4)$$

$$\mathcal{L}_{bg} = \frac{1}{\sum B} \sum |I_{edit}B - I_{src}B|. \qquad (5)$$

**Probability Loss.** In most cases, a single style code channel controls the relevant characteristics of only one attribute. Thus the corresponding control probabilities for each channel computed from $\mathcal{M} \in \mathbb{R}^{(n+1) \times s}$ are expected to be concentrated on a particular attribute, leading to a probability close to 1 for the most related attribute and close to 0 for the rest attributes. In order to encourage the focus, we apply the probability loss as:

$$\mathcal{L}_{prob} = \frac{1}{n} \sum_{i=0}^{n-1} |1 - MAX(Softmax(\mathcal{M}^i))|. \qquad (6)$$

**Optimization.** The global learning objective is a balance of different loss functions with the respective loss weights:

$$\mathcal{L} = \lambda_{attr}\mathcal{L}_{attr} + \lambda_{bg}\mathcal{L}_{bg} + \lambda_{prob}\mathcal{L}_{prob}. \qquad (7)$$

To optimize the above objective loss, in each iteration during training, we randomly sample a source style code $s_{src}$, a reference style code $s_{ref}$, and a target set of attributes $\Omega$ to be transferred.

## 4 IMPLEMENTATION DETAILS

All experiments are performed on a single GeForce RTX4090Ti GPU, with a batch size of 1 for EG3D, and 4 for StyleGANv2. We optimize the network in 300k iterations and utilize Ranger as the optimizer with an initial learning rate of 0.01. For training dataset, we pre-sample 6000 parameter pairs $[z, \mathbf{P}]$ and generate corresponding style codes $s$, filtering out the faces with hats and ensuring the percentage of faces with glasses exceeds 40%. The $s_{src}$ and $s_{ref}$ in each iteration are sampled from the pre-generated style codes, and the size of $\Omega$ is set to 1. We pre-select 3500 style code channels for attribute "Hairstyle" with $d_{Hairstyle} = 1.8$. The loss weights are assigned as $\lambda_{attr} = 3.5$, $\lambda_{prob} = 0.1$, and $\lambda_{bg} = 40$. Please refer to our supplementary material for the detailed descriptor group definition for attribute transfer tasks.

## 5 RESULTS

We provide 3D-aware and 2D attribute transfer results, and conduct ablation study on the crucial parts. For more experimental results, please refer to our supplementary material.

| source image | PREIM3D [24] | reference image (a) | SAT3D | reference image (b) | SAT3D |
|---|---|---|---|---|---|

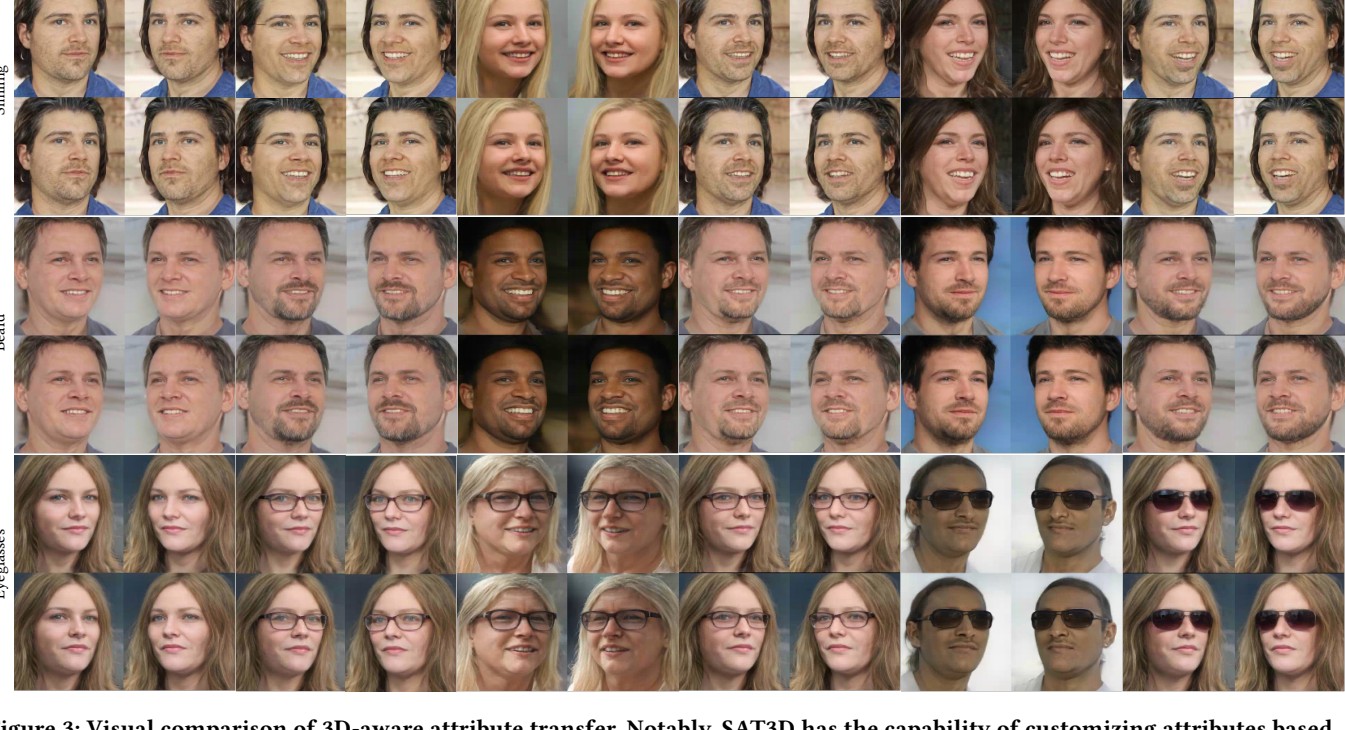
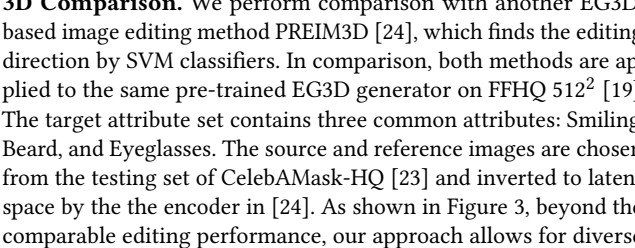

**Figure 3: Visual comparison of 3D-aware attribute transfer. Notably, SAT3D has the capability of customizing attributes based on reference images in addition to competitive editing results.**

## 5.1 3D-aware Attribute Transfer

We apply our method on the pre-trained 3D-aware EG3D generator, which renders multi-view consistent images for each sample and provides comprehensive representation for users. For EG3D pre-trained on facial dataset, there are 6 target attributes that can be transferred, i.e., Hairstyle, Hair color, Eye region, Expression, Beard, and Eyeglasses. The facial segmentation network employed for background loss is BiSeNet [40].

**3D Comparison.** We perform comparison with another EG3D-based image editing method PREIM3D [24], which finds the editing direction by SVM classifiers. In comparison, both methods are applied to the same pre-trained EG3D generator on FFHQ $512^2$ [19]. The target attribute set contains three common attributes: Smiling, Beard, and Eyeglasses. The source and reference images are chosen from the testing set of CelebAMask-HQ [23] and inverted to latent space by the the encoder in [24]. As shown in Figure 3, beyond the comparable editing performance, our approach allows for diverse customization based on reference images.

**Facial Attribute Transfer.** For a more comprehensive presentation, we apply SAT3D to the EG3D generator pre-trained with rebalanced FFHQ [5], which produces better results. We present two editing samples for each attribute in Figure 4, with the source-reference image pairs sampled from latent space. Note the attribute similarity of our editing results to the reference images.

**Generalization to Other Domains.** To demonstrate the generalization ability of SAT3D to different domains, we apply our method to the EG3D generators pre-trained on AFHQv2 Cats $512^2$ [9] and ShapeNet Car $128^2$ [7], and the results are reported in Figure 5. In these domains, there are fewer attributes for editing compared with human faces. The ShapeNet Car mainly can edit color and shape while cat mainly can be edited on fur. We utilize DeepLabv3 [8] to segment the cat images, while no segmentation or background loss is applied to the cars in low image resolution.

**Multi-attribute Transfer.** Sequential editing of multiple attributes can also be achieved, and Figure 6 illustrates the progressive migration of facial attributes from the source image to reference image. As we can observe, at each step, the target attribute is transferred while irrelevant attributes are maintained, and the obtained final face is quite close to the reference image.

## 5.2 2D Attribute Transfer

Our method is also well suited for 2D StyleGAN-based generators. We perform visual comparisons with two classic semantic-guided image editing methods: StyleCLIP [28] and InterFaceGAN [32], which both manipulate latent codes in the latent space of pre-trained StyleGANv2 generator without fine-tuning. For fair comparison, we apply our method on the same generator, which is pre-trained on FFHQ $1024^2$ [19]. The target attribute set contains three attributes that can be edited by all methods: Smiling, Beard, and Eyeglasses. StyleCLIP and InterFaceGAN indicate the presence or absence of attributes by text-pairs and decision boundary respectively, while our proposed SAT3D takes reference images for

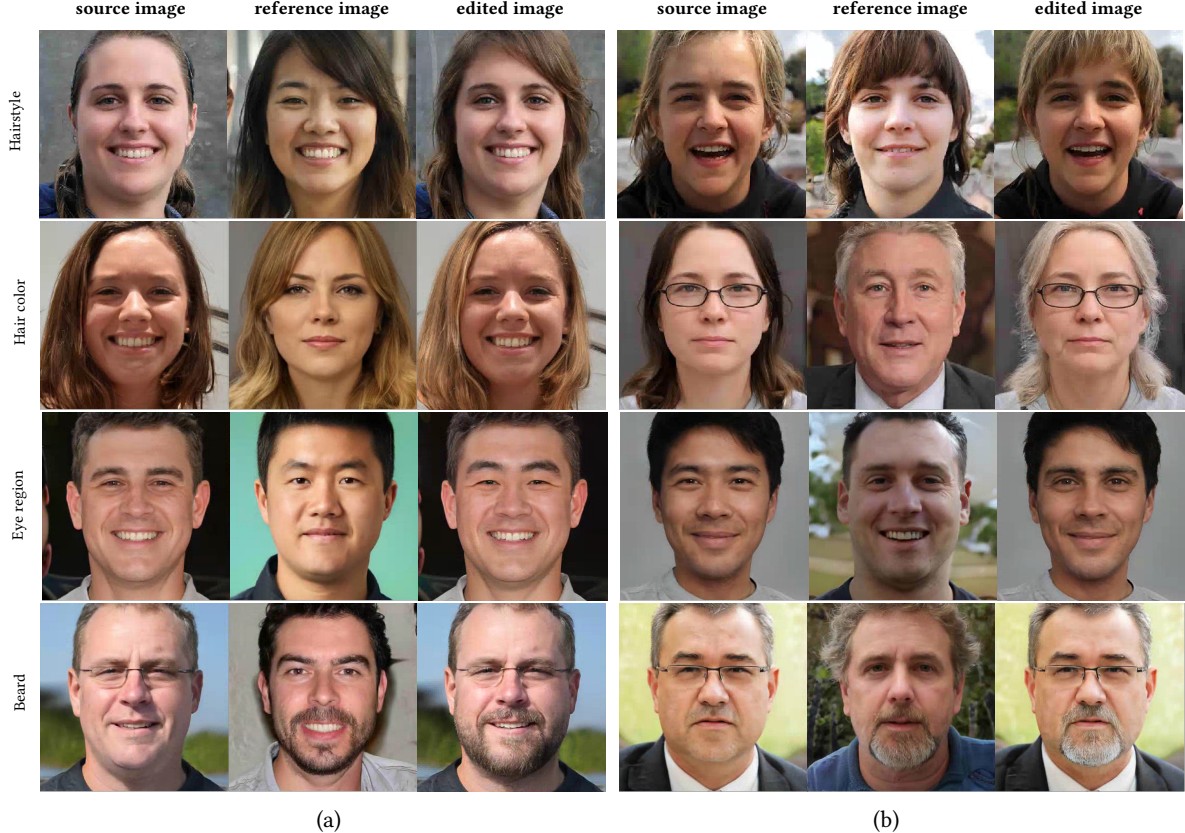

Figure 4: Visualization of attribute transfer on the EG3D generator pre-trained on FFHQ dataset.

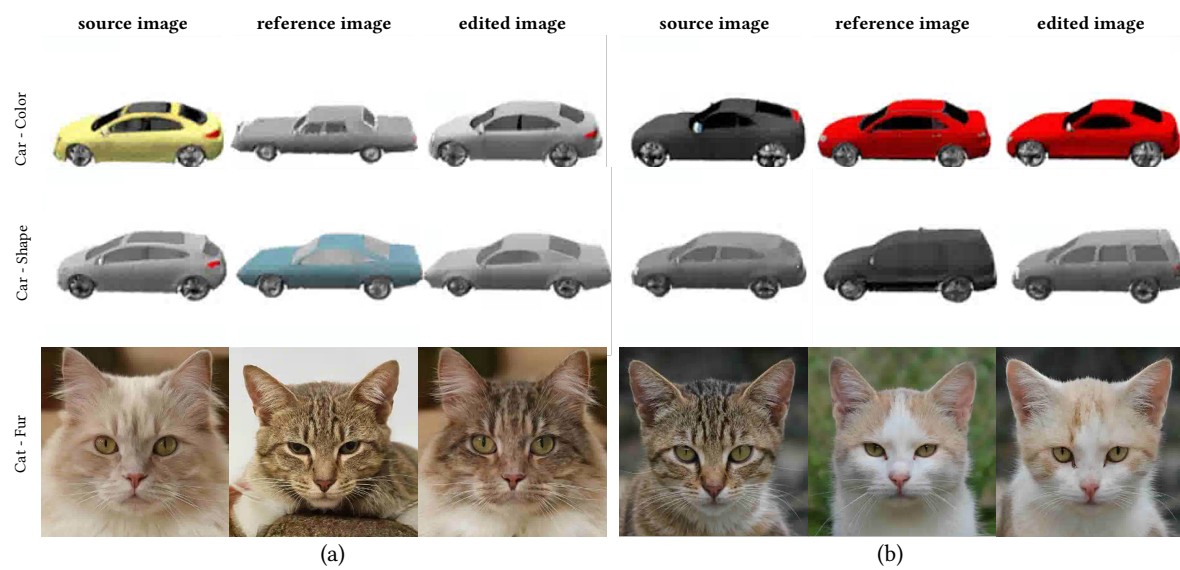

Figure 5: Extra visualization of attribute transfer on the EG3D generator pre-trained on AFHQv2 Cats $512^2$ and ShapeNet Car $128^2$ respectively.

**source image** **+hairstyle** **+hair color** **+eye region** **+expression** **+beard** **+glasses** **reference image**

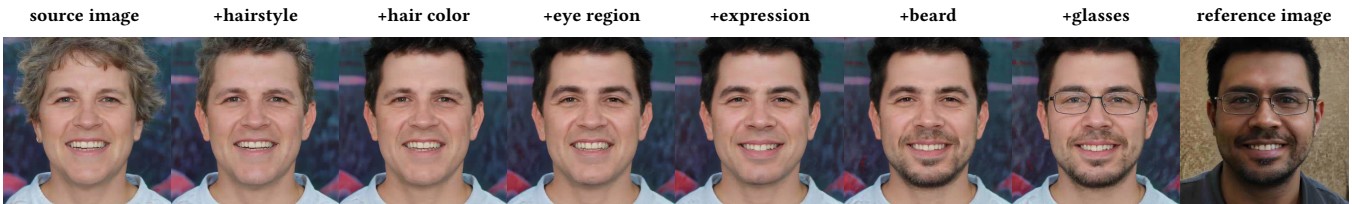

Figure 6: Multi-attribute transfer results.

**source image** **StyleCLIP [28]** **InterFaceGAN [32]** **reference image (a)** **SAT3D** **reference image (b)** **SAT3D**

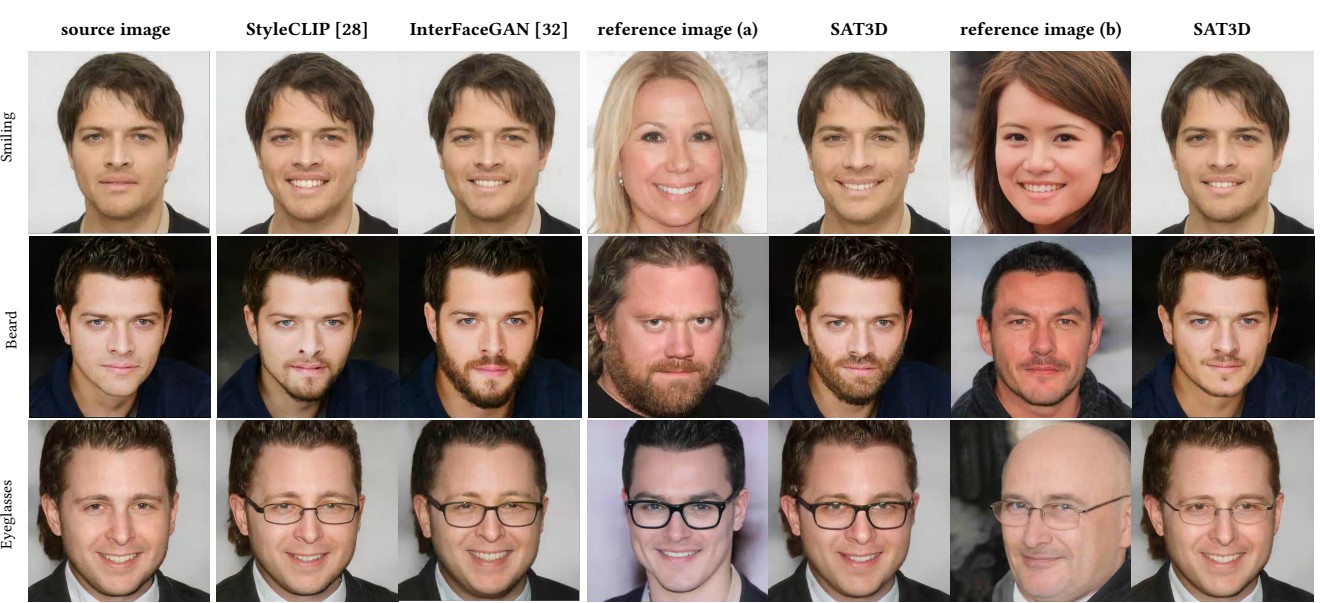

Figure 7: Visual comparison of 2D attribute transfer in the positive direction.

**source image** **StyleCLIP** **InterFaceGAN** **SAT3D**

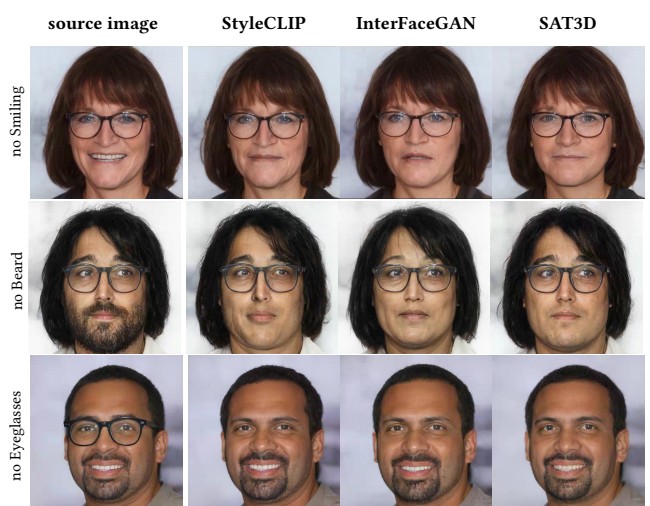

Figure 8: Visual comparison of 2D attribute transfer in the negative direction. We select an image without the target attribute as the reference image for SAT3D.

**source image** **reference image** **SAT3D** **SAT3D w/o $\mathcal{L}_{bg}$**

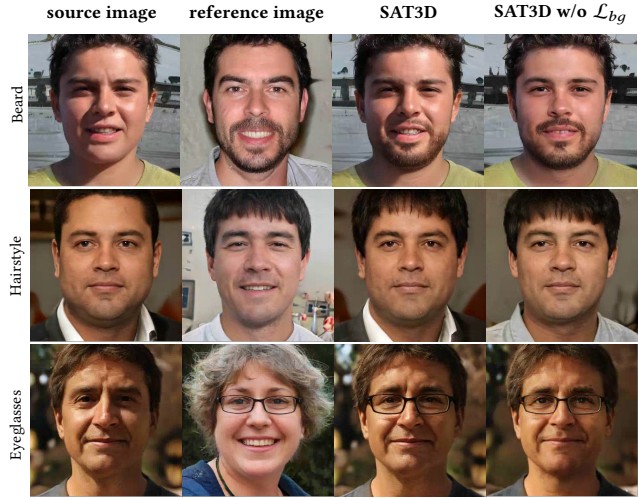

Figure 9: Effect of background loss in training. The background loss is crucial for preserving the lighting, clothes and background textures.

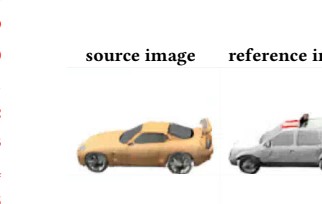

Figure 10: Effect of editing intensity $\delta$. The proper value of $\delta$ varies for each source-reference image pair with roughly in the range of [1.0, 2.25].

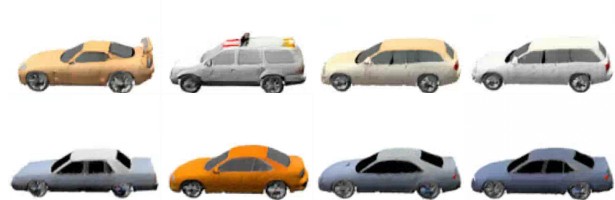

Figure 11: Effect of attribute preserving loss in training. The examplified target attribute is Shape, while Color is expected to be preserved.

guidance. All of the source and reference images are selected from the testing set of CelebAMask-HQ [23] dataset, and inverted to latent space using e4e [35]. In every task, we select the best edited image from a range of intensities for all methods.

In the experiments, three attributes are selected. For each attribute, there are two holistic editing directions, i.e. positive for the presence while negative for the absence (e.g. "Eyeglasses" and "no Eyeglasses"). We conduct comparisons for both situations, as displayed in Figure 7 and Figure 8 respectively. In the positive direction, we select two different reference images each time for SAT3D, demonstrating the ability of our SAT3D to customize attribute characteristics and generate diverse results, which are not supported by other methods. For negative cases, SAT3D simply takes an image without the target attribute as reference. We can observe that SAT3D performs still better than the comparisons at cases "no Smiling" and "no Beard".

## 5.3 Ablation Study

**Background Loss.** The background loss is crucial in preventing uninterested regions from being altered. As presented in Figure 9, in the absence of background loss during training, the target attributes can be transferred with facial structures roughly maintained. However, the background, clothing, shadows, and irrelevant attributes on the face that beyond description are unexpectedly impacted. Adding suppression of unfocused regions can minimize these variations significantly.

**Editing Intensity.** The completion of attribute transfer and the quality of generated images varies with editing intensity $\delta$. As exemplified in Figure 10, for "Eyeglasses", $\delta$=1 gives effective attribute transfer result and there is a gradual transition from the emergence of a rectangular frame to the appearance of sunglasses with $\delta$ increasing. While for "Expression", the characteristics in edited images are not consistent with that of the reference image at $\delta$=1, but rather at $\delta$=2.25. The proper value of $\delta$ differs for each source-reference image pair, which is roughly within range [1.0, 2.25].

**Attribute Preservation Loss.** The attribute preservation loss is intended to retain the characteristics of uninterested attributes within the same region. In practice, we notice that due to the natural properties of gradient back-propagation, using our attribute transfer loss alone already can achieve the goal of exclusively migrating target attribute characteristics in most cases and leaving other attributes in the same region unaffected. Therefore, we do not need to define a comprehensive attribute set for each region, but rather concentrate on the specific attributes of interest. However, at some cases, the entanglement between different attributes still occurs. Taking the car domain as an example in Figure 11, colors are also deviated when we migrate vehicle shapes. Hence, it is necessary to add attribute preservation loss to suppresses the alteration.

## 6 CONCLUSION

For the image-driven semantic attribute editing task, we develop our SAT3D, which transfers semantic attributes of reference images onto the source image. In the highly disentangled style space of pre-trained StyleGAN-based generative models, we explore the correlations between semantic attributes and style code channels. Specifically, a set of descriptor groups are defined for each attribute, and then the attribute characteristics in images are quantitatively measured with QMM utilizing the image-text relevancy evaluation by CLIP models. With the quantitative measurements, we design the attribute loss functions to guide the migration of target attribute and the preservation of irrelevant attributes during training. The transfer results on 3D-aware generators across multiple domains and the comparisons with classical 2D image editing methods demonstrate the effectiveness and customizability of our SAT3D.

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
