# OpenReview forum: "SAT3D: Image-driven Semantic Attribute Transfer in 3D"
_acmmm.org/ACMMM/2024/Conference — MM2024 Poster_

### Official Review · Reviewer_k5kr · 2024-05-03

**Rating:** 3
**Confidence:** 2

**Summary:**

This paper targets image attribute editing. To achieve this, the authors introduce an image-driven semantic attribute transfer method in 3D and present a Quantitative Measurement Module (QMM) to quantitatively describe the attribute characteristics in images and design specific training losses for target attribute transfer and irrelevant attribute preservation. Experiments show the effectiveness of the proposed method.

**Strengths:**

- incorporates a QMM that quantitatively describes attribute characteristics in images using descriptor groups. This module leverages the image-text comprehension capabilities of the CLIP model to guide attribute similarity calculations between images.

- demonstrates effectiveness across multiple domains, including facial attributes, car shapes and colors, and cat fur, showcasing its versatility.

- perform sequential editing of multiple attributes, progressively migrating facial attributes from the source image to the reference image while maintaining irrelevant attributes.

**Limitations:**

- In Figure 3, the authors only compared with PREIM3D, why? Is this the SOTA?
- Why did the authors not provide the quantitative results?

**Suitability:**

3

---

### Official Review · Reviewer_NH3K · 2024-05-09

**Rating:** 4
**Confidence:** 4

**Summary:**

This paper proposes SAT3D, a novel method that transfers semantic attributes of reference images onto the source image in facial image generation field. It explores the correlations between semantic attributes and style code channels in the pre-trained stylegan-based generative model. Experimental results demonstrate the effectiveness and customizability of the proposed SAT3D.

**Strengths:**

1. Experimental results show that the attributes are well decoupled and the proposed method is capable of editing face images with a specified attribute while keeping the others unchanged.
2. The proposed method is able to transfer the style of the specified attribute, e.g. the beard length, the hair style, etc.
3. This paper is well organized and easy to read.

**Limitations:**

1. The type of attributes seems to be fixed, rather than being able to be expressed arbitrarily like text, which might have certain disadvantages in application. This method will be more valuable if the authors can specify attributes through text or some other means to facilitate users to specify the editing area.
2. This method strictly relies on the pre-training ability of stylegan in human-face field. It is not sure whether the proposed method still performs well in other domains, such as anime faces.

**Suitability:**

3

---

### Official Review · Reviewer_YNRn · 2024-05-24

**Rating:** 3
**Confidence:** 3

**Summary:**

This paper proposes an “image-driven Semantic Attribute Transfer method in 3D (SAT3D) based on the pre-trained 3D-aware generator. To address the image-driven semantic attribute transfer task, the author develops a Quantitative Measurement Module (QMM) to quantitatively describe the attribute characteristics in images, and design specific training losses for target attribute transfer and irrelevant attributes preservation.

**Strengths:**

1. This is the first work for image-driven semantic attribute transfer task in 3D.
2. For training guidance, this paper develops a Quantitative Measurement Module (QMM) to measure the attribute characteristics in images, utilizing the zero-shot prediction capability of CLIP.

**Limitations:**

1. The standard quantitative results are missing, such as FID. It is hard to evaluate the visual quality of the edited images.
2. CLIP model is used in the proposed pipeline. Authors should also present the CLIP scores as one of the quantitative metrics.
2. Authors should include more analysis or comparison with the diffusion-based methods, which already demonstrate great ability in attribute transfer. It would be good to see if the proposed module can work in diffusion.

**Suitability:**

3

---

### Meta-Review · Area_Chair_SbCS · 2024-07-01

**Recommendation:** Accept (Poster)
**Confidence:** 5

**Metareview:**

The authors did a good rebuttal. The reviewers unanimously recommend acceptance in the final rating. After checking the rebuttal, the review, and the paper, the AC agrees with this assessment.